# Application of Fluids in Supercritical Conditions in the Polymer Industry

**DOI:** 10.3390/polym13050729

**Published:** 2021-02-27

**Authors:** Karol Tutek, Anna Masek, Anna Kosmalska, Stefan Cichosz

**Affiliations:** Institute of Polymer and Dye Technology, Faculty of Chemistry, Lodz University of Technology, Stefanowskiego 12/16, 90-924 Lodz, Poland; anna.masek@p.lodz (K.T.); anna.kosmalska@p.lodz.pl (A.K.); stefan.cichosz@p.lodz.pl (S.C.)

**Keywords:** supercritical fluid, extraction, particle formation, encapsulation, micronization, polymerization, polymer, foaming, impregnation

## Abstract

This article reviews the use of fluids under supercritical conditions in processes related to the modern and innovative polymer industry. The most important processes using supercritical fluids are: extraction, particle formation, micronization, encapsulation, impregnation, polymerization and foaming. This review article briefly describes and characterizes the individual processes, with a focus on extraction, micronization, particle formation and encapsulation. The methods mentioned focus on modifications in the scope of conducting processes in a more ecological manner and showing higher quality efficiency. Nowadays, due to the growing trend of ecological solutions in the chemical industry, we see more and more advanced technological solutions. Less toxic fluids under supercritical conditions can be used as an ecological alternative to organic solvents widely used in the polymer industry. The use of supercritical conditions to conduct these processes creates new opportunities for obtaining materials and products with specialized applications, in particular in the medical, pharmacological, cosmetic and food industries, based on substances of natural sources. The considerations contained in this article are intended to increase the awareness of the need to change the existing techniques. In particular, the importance of using supercritical fluids in more industrial methods and for the development of already known processes, as well as creating new solutions with their use, should be emphasized.

## 1. Introduction

The supercritical state of chemicals has been known for less than two centuries. In 1822, Charles de la Tour discovered a different “state” of matter besides solids, liquids and gases. Liquids above the critical value of temperature and pressure (called the critical point) exceed the limit beyond which their properties, such as viscosity, density, dielectric permittivity, are the results of both of these states of aggregation. However, they do not undergo any processes of evaporation, condensation or sublimation; i.e., phase transitions. The only known naturally occurring supercritical substance is the water found in hydrothermal vents, where there is increased temperature due to the proximity of magma and increased pressure due to the limited capacity of underground reservoirs.

The liquid state is characterized by rather strong intermolecular interactions, weak enough to displace them. This is also due to an increase in the internal energy of the molecules. There is a certain balance between the internal energy and intermolecular interactions. As a result, we see slight changes in the volume of the liquid with the temperature change, as well as the possibility of taking the shape of the vessel, the characteristic viscosity and the surface tension. On the other hand, the gaseous state is the result of an increase in internal energy and its dominance over the intensity of intermolecular interactions. It is also for this reason that the clear free movement of gas molecules throughout the volume of the vessel is possible. Among the properties, it should also be mentioned that the gas can take the shape of a vessel, and the gas volume also depends on its temperature (assuming a constant pressure). It is possible to expand and compress gases. The important thing is that temperature determines the average amount of internal energy per molecule in a given phase, and pressure is responsible for the probability of these molecules colliding, intermolecular interactions and the distance between them. Increasing the temperature in the system makes the molecules that have more energy detach from each other more easily, and the increase in pressure results in a decrease in the distance between the molecules and their more frequent collisions, which in the next step leads to energy transfer between them. Both these parameters are somewhat antagonistic when looking at the effects of their changes (Figure 1). What the phase is at a given moment depends on their value. Previously, for technical reasons, mainly processes in the liquid phase and less frequently in the gas phase were used. It happened because the reaction was easy to carry out and because of the knowledge of the processes leading to it. However, the discovery of Charles de la Tour and the development of knowledge about the phenomena occurring in supercritical conditions by successive scientists made it possible to use supercritical fluids not only as a laboratory curiosity but also as practically and industrially important processes, thanks to which it is possible to obtain products in an efficient, qualitative and, importantly, profitable manner.

Thermodynamic changes cause changes in the properties of substances. The density, viscosity and diffusivity vary significantly. In the case of supercritical fluids, it is visible that the values of these parameters are intermediate between the liquid and gas phases, which can be seen in the Table 1 below.

An important factor in selecting fluids for supercritical conditions is the relatively low pressure and temperature reached at the critical point, which makes the process cost-effective. Unfortunately, few fluids are capable of becoming supercritical under these conditions. The mainly used supercritical solvents are (Table 2):

One of the most widely used supercritical solvents in the research and industry is CO_2_ due to its favorable process conditions, such as room temperature and relatively low critical pressure compared to other popular solvents. In addition, supercritical CO_2_ is characterized by nontoxicity, high volatility, a wide change in density, etc. Depending on temperature and pressure, the dielectric constant changes from 1.02 to 1.68, according to Michaels et al. [19]. The polarizability value is 27.6 × 10^−25^ cm^3^ [18], and the dipole moment is equal to 0,0 due to the symmetry of the structure of the carbon dioxide molecule.

In recent years, there has been a growing interest among researchers in the use of supercritical fluids in various industries, including one of the most important in terms of environmental impact, which is the polymer industry. For this reason, it is important to develop modern and environmentally friendly basic chemical processes [20].

## 2. Supercritical Fluid Extraction (SFE)

At present, due to the growing trend of ecological solutions in the chemical industry, more and more advanced technological solutions are being implemented. An important element of the chemical industry is obtaining natural compounds from herbs, fruits, vegetables [21], and a whole range of compounds found in fauna and flora is indicated in the studies by Reverchon et al. and Cappuzo et al. [1,2,5,13,15,22,23]. These compounds are used in each of the possible industries, but the most important of them today are the cosmetics [24], medical [8] and food industries [25]. However, extracts as functional additives are also gaining growing popularity in the polymer industry [14,26,27]. 

One of the most important applications of supercritical fluids is extraction (Figure 2). Until now, the most important of the methods were based on liquid extraction, where parts of plants were immersed in water, alcohol or hydrocarbons, and this is where important and desired natural compounds were dissolved. Unfortunately, apart from them, other unwanted ones were often obtained, which had to be removed in the next stages. Another problem was that the solvent always remained in the obtained extract, at least to a trace degree. Enzymatic extraction followed by lyophilization was another important method. However, one of the best methods currently in use is extraction in supercritical solvents, according to reviews da Silva et al. [28].

The equipment used in SFE is unfortunately expensive, as it has to withstand and handle high pressures and temperatures. For this reason, it is not very popular in laboratories. 

The raw material intended for extraction is placed in a pressure extractor. A solvent is then supplied to the system, heated by the action of a heat exchanger and compressed by means of a compressor to achieve supercritical conditions suitable for the extraction of a given group of natural compounds. The as-obtained solvent in the supercritical phase is directed to the extractor, where it diffuses through the raw material in order to extract the desired extracted component. In the next stage, the solvent solution with the extract goes to the separating funnel, where, as a result of depressurization and temperature changes, the solvent turns into a gas. However, the pressure and temperature variation in the system is for the most part. The solvent can be recycled for reuse by additional recirculation. As a result of the aforementioned separation, a precipitated extract of solvent-free natural compounds is obtained. Supercritical CO_2_ is a good solvent for a large group of nonpolar small-molecule compounds, and unfortunately, it does not dissolve most polymers under mild conditions. The exceptions are amorphous silicones and fluoropolymers with an amorphous structure. SC-CO_2_ has a high quadrupole moment value, the interactions of which occur at a shorter distance from polar interactions. Importantly, as the pressure increases, the density of supercritical carbon dioxide increases. Unfortunately, with increasing pressure and thus density, the diffusivity of this supercritical fluid will decrease. As a result, the process temperature is increased to increase the dissolving potential. By changing the temperature and pressure of the flowing solvent, the extraction conditions can be controlled to selectively obtain the desired compounds, which was indicated in the study by Hedrick et al. [29].

Various solvents are used in the SFE technique. However, the cost-effectiveness of the method depends on the parameters in which they become supercritical [6]. In this case, carbon dioxide is an excellent medium for nonpolar or low-polarity compounds due to the fact that already at 30 °C, and at a pressure of just over 7 MPa, it becomes supercritical. An important factor is that by increasing the pressure, the diffusivity and density can be controlled, and it is also possible to change the extracted substances from nonpolar to those of low polarity. In order to extract natural compounds with greater polarity, there are two possibilities: either add another more polar cosolvent to CO_2_, such as methanol, or completely change the dissolving medium to polar [30]. The solvents still popular in the SFE method include: methanol, diethyl ether, ammonia and acetone.

However, all of these solvents are unfortunately characterized by a higher temperature or pressure at which they become supercritical. Carbon dioxide is a preferred extractant because the temperature at which the supercritical state is obtained is slightly higher than room temperature, which means that most biologically active compounds will be stable under these conditions and will not decompose (Table 3). An additional advantage is that this type of extraction is carried out under anaerobic conditions, thanks to which the natural compounds obtained by us will not be oxidized. Inorganic compounds are also not extracted, which is important in particular in the extraction from marine plants and animals, which are often contaminated with heavy metals. Another and no less important advantage of the use of CO_2_ is that it is a simple compound that commonly occurs around us in nature, and it is not harmful to us after consumption or contact with the skin, which is extremely important in the subsequent use of extracts in the cosmetics, pharmaceutical and food industries [31]. It belongs to the so-called “green chemistry” solvents, which do not cause excessively harmful effects on the environment, as emphasized by Majewska et al. [32]. Carbon dioxide is used to obtain in the SFE method such groups of compounds as flavonoids, anthocyanins and polyunsaturated fats.

The main advantages of supercritical extraction include [38]:Operation at low temperatures, thanks to which chemical compounds that are not thermally resistant are not degraded.Use of nontoxic solvents, which allows for a more ecological approach to technological processes.The ability to selectively regulate solubility through changes in pressure and temperature increases the selectivity of a chemical reaction.Complete separation of the solvent from the extract reduces the contamination of products with them, as a result of which it is possible to make products intended for direct contact with humans.The possibility of recirculation of the solvent in the system, which reduces the costs of the process and lowers the process costsFractionation of the compounds obtained by extraction during their isolation, so it is possible to obtain only the desired product.The extraction is carried out under anaerobic conditions, which prevents the oxidation of valuable natural substances.As for disadvantages, we can mention:A high cost of building the installation due to the need to withstand very high pressure.Considerable energy expenditure for solvent compression and heating.Incomplete knowledge of the SFE process and frequent empirical determinations, which results from previous low interest in this technology and the presence of a small number of research installations.

## 3. Particle Formation, Micronization and Encapsulation

The main methods of producing microcapsules were:The emulsion method, consisting of obtaining an emulsion of immiscible liquids containing substances forming a shell and a core and then removing the solvent.Spray drying, which is based on dissolving the shell in a solvent and then dissolving, suspending or emulsifying the filling substance into this solution. The next stage is spraying the liquid through atomizers or nozzles, which causes the solvent to evaporate and the active substance to be deposited on the shell.Extrusion, which is used primarily for the production of microcapsules of oils in a carbohydrate matrix. Included here are three techniques: melt injection, melt extrusion and centrifugal extrusion.Coacervation, consisting of separating the phases in a solution of colloids or polymers and creating at least two liquid phases. The course of the coacervation process begins with phase separation in the polymer solution under the influence of temperature, pH or the addition of salt or an incompatible polymer. In this way, coacervate droplets are produced, which are adsorbed onto the surface of the active substance, thus forming the capsule shell. Adjusting the concentration of added salt, viscosity and molecular weight of the polymer allows controlling the size of the microcapsules obtainedPolymerization in situ, which is based on the simultaneous occurrence of the shell polymerization process and the surrounding of the active substance with the produced polymer. In situ polymerization takes place without the addition of reactive agents. This process often produces capsules based on a melamine-formaldehyde film formed by the reaction of melamine with formaldehyde on the surface of an oil droplet. These casings are characterized by high strength and stability. The process of producing microcapsules by the in situ method consists of the production of melamine-formaldehyde precondensate and its prepolymerization, followed by adding oil to the solution and its emulsification. Subsequently, the temperature of the emulsion is increased, which causes the prepolymer to polymerize and form a microcapsule shell around the active ingredient.Lyophilization, where substances subjected to lyophilization, such as oils, are dissolved in water, and then, by reducing the pressure, the water is removed from the system, passing directly to the gaseous state. This method retains the maximum amount of volatile compounds.

The processes of particle formation, micronization and encapsulation mainly use supercritical fluids [39,40,41,42,43,44,45]. Despite their differences, they use the same thermodynamic basis and manufacturing methods to carry out these processes. The supercritical fluid is used either, as a solvent as in the rapid expansion of the supercritical solution (RESS) method or as a solute that produces particles from a gas-saturated solution (PGSS). However, there is still the option of carrying out the process in which the supercritical liquid is an antisolvent, thanks to which the precipitation of new particles (supercritical antisolvent (SAS)) is induced [46,47,48,49,50,51,52,53]. These are methods increasingly used in the industry because they allow for very good control of the size of the particles formed and for the entrapment of active substances in them, which are sensitive to various reaction conditions, such as high temperatures or the action of certain solvents. In order to be able to fully understand and control the above processes, it is important to have a detailed knowledge of the thermodynamic formation of phase equilibria in mixtures. For this reason, they are not widely used on a large scale in the industry, as research in these directions is still being developed. Only the pharmaceutical industry in drug development, the food and cosmetics industries will make the greatest contribution in this direction so far [10,54,55,56,57,58,59,60,61,62,63,64].

### 3.1. Rapid Expansion of Supercritical Solution (RESS)

One of the primary methods used is the rapid expansion of supercritical solution (RESS) (Figure 3). It was first used in a patent by Smith and Walsh in 1984. In the present process, the supercritical fluid is a solvent or carrier of the entrapped substance. The second case is more common because the active substance dissolves with the material that later forms the coating in the supercritical fluid. In the next step, the coprecipitation of the substances is carried out by spraying the mixture under atmospheric pressure. In this case, the solvent, which under supercritical conditions was a condensed fluid, turns into a gaseous state, which is no longer able to dissolve the substances contained in it. As a result of this transition, there is a rapid reduction in density and solubility, and consequently, the precipitation of particles occurs.

Unfortunately, this method has some limitations in use, as few substances have good solubility in the most commonly used supercritical CO_2_. Additionally, the active substance forms the core and the substance forming the shell must be soluble in this solvent. To circumvent these problems, other supercritical organic solvents are used or a small amount is added as a cosolvent. In addition, a big problem is the difficult control over the morphology and the amount of the active substance closed. This problem was avoided by the supercritical precipitation of the carrier onto the previously prepared capsules of the active substance, most often using a fluidized bed [65,66,67,68,69,70,71].

### 3.2. Supercritical Antisolvent (SAS)

Micronization with the use of a supercritical fluid as an antisolvent has several variants of modification. The basic variant that is also used in many companies is SAS (Figure 4). This process is based on the assumption that the solubility of the previously dissolved substance in the primary solvent decreases with an increase in the amount of the supercritical fluid in the mixture. The formation of microcapsules takes place through crystallization due to the reaction between the primary solvent (usually organic) and the antisolvent [72]. There, the process of diffusion of the antisolvent to the organic solvent phase takes place, and then the evaporation of the organic solvent into the antisolvent. The SAS process is limited by the ability to separate the resulting solids from the solvents used. The size and shape of the obtained particles depend on the type of core- and shell-forming substances used, process parameters and the percentage and separation of the solvent and antisolvent.

### 3.3. Aerosol Solvent Extraction System (ASES)

This method is an improvement over the basic SAS method (Figure 5). It is based on spraying a solution of the active ingredient through a nozzle into a reservoir containing the supercritical fluid. During the atomization and dissolution of the supercritical fluid in the liquid droplets, a large increase in volume occurs, and at the same time, the dissolving capacity of the liquid solvent decreases. This causes supersaturation to rise dramatically, resulting in the precipitation of small particles of similar size [10].

### 3.4. Particles from Gas-Saturated Solution (PGSS)

This is a process in which active substances do not have to be soluble in the supercritical fluid; however, they must absorb it in significant amounts, even up to 40%. In an initial step, the supercritical fluid and the solutions of the shell-forming substance and the active substance in the core of the future microcapsules are mixed. In the next step, the previously obtained mixture is expanded to atmospheric pressure by means of a nozzle. As a result of the expansion, the supercritical fluid expands and evaporates, increasing the atomization effect and quickly picks up excess heat. All this allows the shells of molecules to solidify almost instantly (Figure 6). 

As a result of the above operations, a powder with micrometric sizes and a controlled size distribution is obtained [73,74,75]. The PGSS process is the most widely used antisolvent method using supercritical conditions in the industry [76]. The process is carried out in relatively mild conditions, thanks to which it is possible to use substances sensitive to high temperatures and oxidation [77,78].

### 3.5. Electrospraying (ESPR)

This is one of the latest modifications to the method of obtaining particles. It uses a conventional method to control good viscosity and surface tension but with fluid support under supercritical conditions. As Baldino and coauthors describe it in his work [79], it consists of spraying the solution through a nozzle, using additionally the potential difference between the nozzle tip and the ground electrode. If a sufficiently strong voltage is obtained to overcome the cohesive forces in the solution, it is possible to produce a haze consisting of micrometric and nanometric particles. If the voltage is too low and the cohesion forces in the solution are not overcome, fibers are produced as a result of a process called electrospinning (ESP). Unfortunately, ESPR has a certain limitation related to very low efficiency. For the process to run properly, the solution flow must be insignificant. An additional difficulty is that the solution must have low viscosity and surface tension. The introduction of sc-CO_2_ in this case allows reducing the cohesive forces in the sprayed solution. An important advantage is that the same apparatus can be used for both electrospray and electrospinning, changing only the process parameters mentioned above.

## 4. Impregnation and Plasticization

The impregnation process is used in industry to apply active substances to the matrix [80,81,82,83,84,85]. The studies appear in environmentally friendly ways of plasticizing polymers [86], and one of them is the use of modification using supercritical fluid conditions. In the case of our considerations, the matrix will be a polymer. The impregnation process is possible when the active substance is well-soluble in the supercritical fluid; the polymer swells in contact with the supercritical fluid, and there is a favorable partition coefficient so as to apply as much of the active substance to the matrix as possible [87,88,89,90,91,92]. The impregnation process is most often used in the medical industry, resulting in drug carriers and other pharmaceuticals [92,93,94,95,96,97,98,99,100,101]. Implementation in the food industry is becoming equally significant [102,103]. In both of these areas, an important advantage of impregnation with supercritical fluids is that the product is free from solvents used in the impregnation process [104,105].

Impregnation processes supported by supercritical fluids can be divided into those in which the most important is the solubility of the solute and affinity of the solute to the polymer matrix.

In the case of the former, it is important to distinguish two submechanisms, the first of which focuses on dissolving the active substance in a supercritical fluid and then depositing it on the polymer matrix after lowering the pressure. Unfortunately, often, the solute has a low affinity for the matrix, and the active substances trapped in it recrystallize without a molecularly dispersed formulation. The second of the mentioned mechanisms uses high coefficients of substance partition between the liquid phase and the polymer matrix to which the active substance has a high affinity. In this case, drugs, dyes and organometallic compounds modified in the polymer matrix are obtained [106,107,108,109,110].

Polymers interact with supercritical fluids not only at temperatures above T_m_ but also with polymers in the glass state. During the interaction, the supercritical fluid is dissolved in the polymer matrix, its swelling and the glass transition temperature is lowered as a result of plasticization [95,111,112]. Thanks to the latter, it allows for easier penetration of the active substance between entangled polymer chains and greater quantitative application in the matrix. The active substance closed in this way has a limited release from the finished product, which has found its practical application in the production of drugs with a controlled release time of active substances. In the process of impregnation, it is very important to select the right components of the matrix, which must be swollen by the supercritical fluid and active substances that must be soluble in it [113,114,115,116,117,118,119].

## 5. Other Processes Using Supercritical Fluids in the Polymer Industry

### 5.1. Foaming

Porous structures are a common material used in the polymer industry. For this reason, research was conducted on the use of supercritical fluids for the production of this type of structure. Of particular interest are foamed biopolymers such as poly(lactic acid) (PLA) or polycaprolactone (PCL), which can be used to create temporary skeletons in tissue engineering in medicine [120,121,122]. This technique uses a polymer that is exposed to fluids under supercritical conditions, which causes plasticization of the polymer matrix and lowers the glass transition and melting temperature. After depressurization, CO_2_ is released, and thermodynamic instability determines the supersaturation of dissolved carbon dioxide in the polymer matrix and cell nucleation takes place, as suggested by Karvanja et al. [123]. Unfortunately, there are limitations to this method as the interaction between supercritical CO_2_ and the polymer matrix is limited to low T_g_ amorphous or semicrystalline regions only [124,125].

One of the greatest advantages of foamed polymers made with the help of supercritical solvents is the very good control of the morphology of the resulting structures, limiting the use of organic solvents, carrying out the processes in mild environmental conditions, which reduces the degradation of biological compounds such as proteins or polysaccharides [126,127,128]. In addition, an important aspect is the reduction of the consumption of environmentally unfriendly compounds; e.g., HCFCs used in the production of porous or flammable materials, such as frequently used hydrocarbons, e.g., pentane, as Di Maio and coauthors indicate in his work [129]. Unfortunately, carrying out the foaming process with liquids under supercritical conditions is a significant challenge. This is due to the overlapping of many different processes and their extensive knowledge in relation to a number of sciences such as thermodynamics, material engineering or process engineering. However, the rapid development of technology and knowledge allows for a better understanding of the process itself, which also translates into better control of the size and characteristics of the cellular structure in porous materials.

### 5.2. Polymerization

Polymerization in supercritical fluids is an interesting alternative to the conventional method of polymerization in organic solvents. Supercritical CO_2_ is easily removed after the polymerization process and, in the case of radical polymerization negligible chain transfer, is observed compared to organic solvents [72,130,131]. Most polymers are insoluble in supercritical carbon dioxide, except for amorphous polysiloxanes and fluoropolymers. In the case of conducting the polymerization process in supercritical conditions, it is easier to control the morphology and moisture of the particles. High diffusivity and high mass transfer, as well as low viscosity of supercritical carbon dioxide, significantly reduce the Trommsdorff effect during polymerization. This limits local overheating and degradation of the already-formed polymer chains. The polymerization process carried out in supercritical conditions is characterized by better kinetics and obtaining polymers with a higher molecular weight than conventionally obtained in organic solvents. Unfortunately, the low solubility of polymers in SCF-CO_2_ requires the use of more stabilizers; however, in order to reduce their amount, it is possible to add small amounts of organic solvents to the supercritical conditions [16,132,133,134,135,136].

## 6. Future Perspectives

While reviewing the available literature on supercritical techniques in the polymer industry, attention should be paid in recent years to the increased interest in polymers such as poly(methyl methacrylate) [137,138,139,140,141], polystyrene, poly(lactic acid) [112,142,143,144,145], polycaprolactone [122,146,147,148] or poly(ethylene oxide) [149,150]. On the other hand, the main processes included polymerization, foaming and the production of films and composites [151]. Research that may be carried out in the future in this field will probably focus on the relationship between complex thermodynamic properties and the behavior of polymers and the conditions of the process, such as temperature, pressure or heat exchange rate. An interesting direction of development is the study of controlled polymerization in supercritical conditions and aging processes of polymer matrices as a result of long-term action of supercritical carbon dioxide. However, due to the small number of centers researching polymer reactions in supercritical conditions, the development of this subject will be slow and will allow for significant innovation. What Nalawade et al. point out in their works is that an extremely important element is still the understanding of the basic relationships between supercritical fluids and polymers, their interactions and process kinetics because the current model achievements of the conditions of temperature, pressure and time dependence are not able to fully describe the experimental experiments in a correct way [11,135,152,153,154,155,156,157,158,159,160]. However, advances in the use of supercritical fluids will be driven by the need to reduce the use of environmentally toxic organic solvents and by the demand for these technologies by the medical [161], pharmaceutical [162], cosmetic and food industries [7,18,20,66,149].

## Figures and Tables

**Figure 1 polymers-13-00729-f001:**
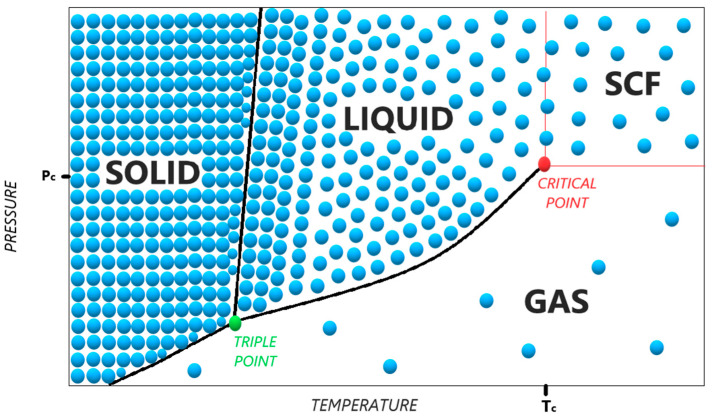
Schematic phase diagram of the dependence of the density changes of the medium depending on the pressure and temperature. Density changes are graphically represented as changes in the blue dot density.

**Figure 2 polymers-13-00729-f002:**
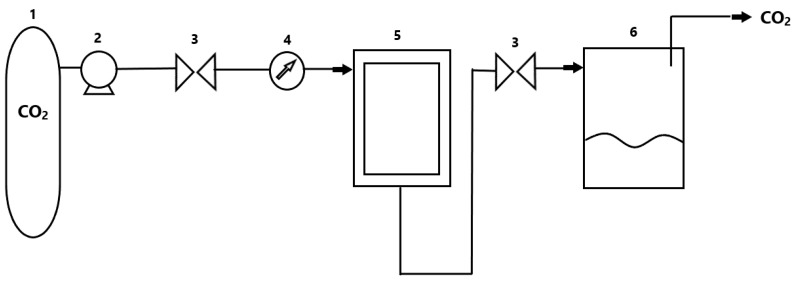
A simplified diagram of the installation for fluids extraction in supercritical conditions. (1) CO_2_ tank; (2) pump; (3) valve; (4) temperature control system; (5) extractor; (6) separator.

**Figure 3 polymers-13-00729-f003:**
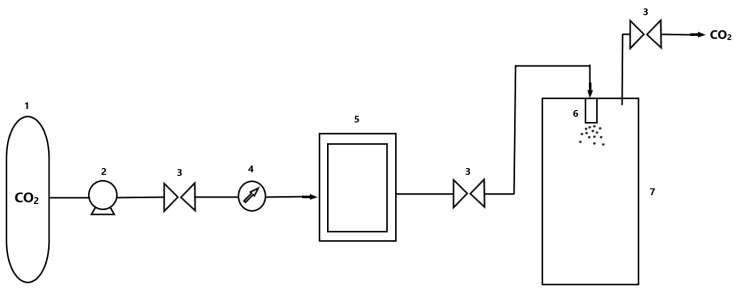
Rapid expansion of supercritical solution (RESS) process installation diagram. Legend: (1) CO_2_ tank; (2) pump; (3) valve; (4) temperature control; (5) tank with agitator; (6) nozzle; (7) separating tank.

**Figure 4 polymers-13-00729-f004:**
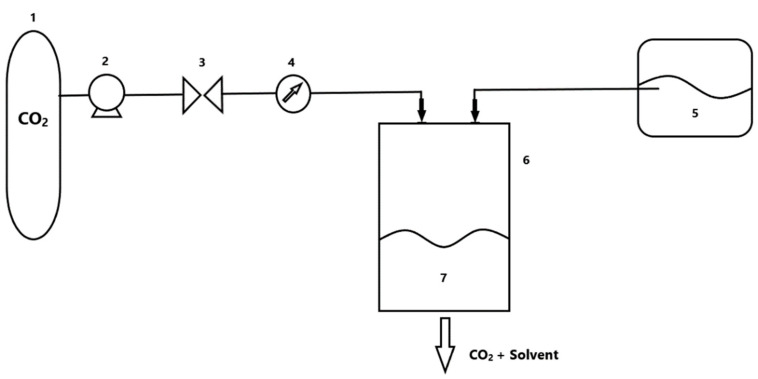
Supercritical antisolvent (SAS) process installation diagram. Legend: (1) CO_2_ tank; (2) pump; (3) valve; (4) temperature control; (5) tank with organic solvent; (6) separating tank; (7) precipitated crystals.

**Figure 5 polymers-13-00729-f005:**
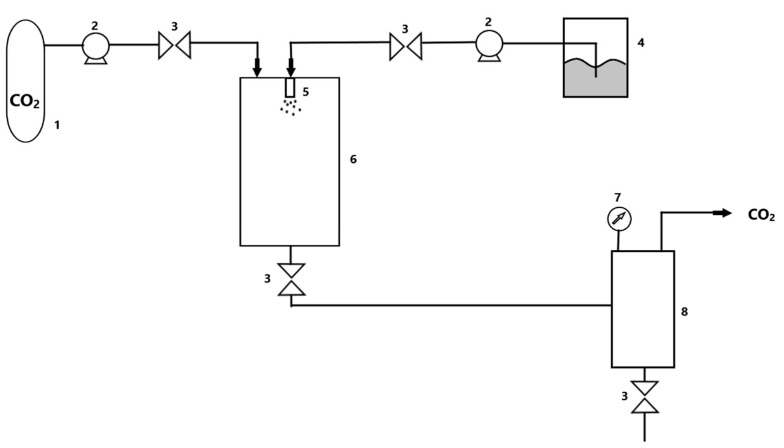
Aerosol solvent extraction system (ASES) process installation diagram. Legend: (1) CO_2_ tank; (2) pump; (3) valve; (4) active substance solution; (5) nozzle; (6) high-pressure tank; (7) temperature control; (8) low-pressure tank.

**Figure 6 polymers-13-00729-f006:**
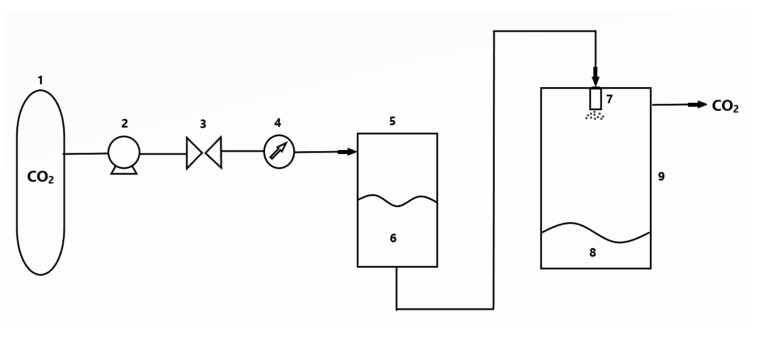
Particles from a gas-saturated solution (PGSS) process installation diagram. Legend: (1) CO_2_ tank; (2) pump; (3) valve; (4) temperature control; (5) static rotor; (6) carrier and active substance mixture; (7) nozzle; (8) powder/product; (9) tank.

**Table 1 polymers-13-00729-t001:** Comparison of physical parameters in different physical states [1,2].

Phase	Density (kg/m^3^)	Viscosity (µPa × s)	Diffusivity (mm^2^/s)
Gas	1	10	1–10
SCF	100–1000	50–100	0.01–0.1
Liquid	1000	500–1000	0.001

**Table 2 polymers-13-00729-t002:** Critical parameters of exemplary solvents.

Solvent	Critical Temperature (K)	Critical Pressure (MPa)	Critical Density (g/cm^3^)	Ref.
Acetone	508.1	4.7	0.278	[3]
Ammonia	405.6	11.3	0.235	[4]
Carbon dioxide	304.2	7.4	0.468	[5]
Diethyl ether	467.6	3.6	0.265	[3]
Methanol	512.6	8.1	0.272	[6]
Toluene	591.7	4.1	0.292	[7]
Water	647.3	22.0	0.322	[8]
Benzene	562.2	4.9	0.304	[9,10,11]
Chlorodifluoromethane	384.9	3.9	0.522	[9,10,12]
Ethane	305.6	4.9	0.212	[9,10,13]
Ethylene	282.5	5.1	0.220	[9,10,14]
n-Propane	367.0	4.3	0.225	[9,10,15]
Cyclohexane	553.3	4.0	0.270	[16,17]
Nitrogen dioxide	309.4	7.2	0.457	[16]
n-Pentane	469.6	3.4	0.232	[16]
Isopropanol	508.6	5.4	0.274	[16]
Methane	190.5	46.4	0.16	[18]
C_2_F_6_	292.9	30.6	0.62	[18]
SF_6_	318.6	37.2	0.73	[18]
Propylene	364.9	46.1	0.24	[18]
Ethanol	516.5	63.8	0.28	[18]
Isobutanol	548.1	43.0	0.27	[18]
Pyridine	647.2	220.5	0.32	[18]

**Table 3 polymers-13-00729-t003:** Examples of groups of active compounds obtained by supercritical extraction.

Name of the Natural Matrix	Functional Compound	Group of Functional Activity	Supercritical Fluids	Conditions of Extraction	Reference
Melissa	Phenol	Antioxidant	CO_2_	100 bar, 35 °C	[33]
Saffron	Volatile Oil	Antimicrobial	CO_2_ and isopropilic alcohol	300 bar, 40 °C	[7]
Sage	Oil	Hipocholesterolemic	CO_2_	250 bar, 60 °C	[34]
Sage	Essential Oil	Antispasmodic	CO_2_	128 bar, 50 °C	[7]
Anis Seed	Triglycerides	Diuretic	CO_2_	250 bar, 40 °C	[35]
Chamomile	Oleoresin	Anti-inflammatory	CO_2_	160 bar, 40 °C	[36]
Clove bud	Essential Oil	Antiseptic	CO_2_	120 bar, 50 °C	[37]
Stevia	Glycosides	Hypoglycemic Hypotensive	CO_2_	200 bar, 30 °C	[7]

## Data Availability

The data presented in this study are available on request from the corresponding author.

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
