# Peer review of "Application of Fluids in Supercritical Conditions in the Polymer Industry"

_polymers, 2021, doi:10.3390/polym13050729_

Round 1

Reviewer 1 Report

This manuscript is well organized, the structure is logical and the discussion and analysis are in-depth, I think, it can be accepted after minor revision.
1.The author should carefully proofread the full text to avoid some minor errors, such as carbon dioxide and CO2 in some places. 
2. Following article should be cited in "Impregnation and plasticization" section.
L Xu, J Zhao, S Qian, X Zhu, J Takahash, Green-plasticized poly(lactic acid)/nanofibrillated cellulose biocomposites with high strength, good toughness and excellent heat resistance. COMPOS. SCI. TECHNOL. 2021, 203, 108613.

Author Response

Institute of Polymer and Dye Technology

Technical University of Lodz

90-924 Lodz, ul Stefanowskiego 12/16, Poland

Tel.: +48 42 631 32 23, Fax: +48 42 636 25 43

February 15, 2021

Polymers

Dear Professor,

We are resubmitting our revised paper entitled Application of fluids in Supercritical Conditions in the Polymer Industry by, Karol Tutek, Anna Masek, Anna Kosmalska and Stefan Cichosz with a request to reconsider it for publication in Polymers.

We have carefully considered the Editor and Reviewers' comments. The manuscript was revised exactly according to these comments. The list of responses to the reviewers’ comments and corrections made in the manuscript is attached.

The manuscript has not been previously published, is not currently submitted for review to any other journal, and will not be submitted elsewhere before a decision is made by this journal.

For correspondence please use the following information:

corresponding author: Anna Masek

Institute of Polymer and Dye Technology

Technical University of Lodz

90-924 Lodz, ul Stefanowskiego 12/16, Poland

Tel.: +48 42 631 32 93

Fax: +48 42 636 25 43

Yours sincerely,

Ph. D., D.Sc. Anna Masek

Answer to reviewer #1 comments

Comments and Suggestions for Authors:

This manuscript is well organized, the structure is logical and the discussion and analysis are in-depth, I think, it can be accepted after minor revision.

1.The author should carefully proofread the full text to avoid some minor errors, such as carbon dioxide and CO2 in some places.

  1. Following article should be cited in "Impregnation and plasticization" section.

L Xu, J Zhao, S Qian, X Zhu, J Takahash, Green-plasticized poly(lactic acid)/nanofibrillated cellulose biocomposites with high strength, good toughness and excellent heat resistance. COMPOS. SCI. TECHNOL. 2021, 203, 108613.

Answer: We are grateful for the Reviewer's suggestions and we tried to meet all expectations. The CO2 errors that were identified and noticed have been corrected. Proposed article "L Xu, J Zhao, S Qian, X Zhu, J Takahash, Green-plasticized poly (lactic acid) / nanofibrillated cellulose biocomposites with high strength, good toughness and excellent heat resistance. COMPOS. SCI. TECHNOL. 2021, 203 , 108613 "was quoted in the indicated section" Impregnation and plasticization ". The linguistic correction indicated in the review was carried out. We would like to thank you for any comments that were helpful in improving the text of the article.

Reviewer 2 Report

From my point of view this review is not thorough enough, and the descriptions of the processes and technologies should be more accurate and detailed. Some statements are not clear and lead to confusion. As an example, authors can find the following very detailed review on very similar topic published in 2019: (Dense CO2 technology: Overview of recent applications for drug processing/ formulation/delivery, Chemical Engineering & Processing: Process Intensification 140 (2019) 64–77, doi.org/10.1016/j.cep.2019.04.009)

Moreover, the following issues should be addressed:

Tittle: When one refers to supercritical conditions, we refer to fluids, not to liquids. (same for Figure 2).

Page 1, line 32: They do not undergo any processes of evaporation, condensation or sublimation. What do you mean? These processes depend on the temperature and pressure conditions.

Page 1, line 35: Is it also possible to find reservoir fluids in supercritical conditions?

Table 1: Fluid, what do you mean by fluid here?

Page 3: Line 85: Check polarization units.

Page 4: Line 123: ‘is the most common’, please elaborate this sentence.

Page 4: Line 125: Can you elaborate more in the fact that the temperature and pressure conditions of the solvent will affect the selectivity of the extraction process? Also some more thorough explanation of the change of the extracted substances from non-polar to polar by increasing pressure. (line 133).

I could not find Table 3 presented in the manuscript.

Page 5, line 162: ‘increases the possibility of a chemical reaction’: Please elaborate more this statement.

SAS: Page 7, line 223: ‘The formation of molecules takes place through crystallization due to the reaction …’. Which reaction are you referring to in the anti solvent process? Which molecules are formed?

Page 8, line 247: What is the future molecule?

Page 9, line 296: Please define the acronyms used.

Page 9, line 302: What do you mean here by cell nucleation?

Author Response

Institute of Polymer and Dye Technology

Technical University of Lodz

90-924 Lodz, ul Stefanowskiego 12/16, Poland

Tel.: +48 42 631 32 23, Fax: +48 42 636 25 43

February 15, 2021

Polymers

Dear Professor,

We are resubmitting our revised paper entitled Application of fluids in Supercritical Conditions in the Polymer Industry by, Karol Tutek, Anna Masek, Anna Kosmalska and Stefan Cichosz with a request to reconsider it for publication in Polymers.

We have carefully considered the Editor and Reviewers' comments. The manuscript was revised exactly according to these comments. The list of responses to the reviewers’ comments and corrections made in the manuscript is attached.

The manuscript has not been previously published, is not currently submitted for review to any other journal, and will not be submitted elsewhere before a decision is made by this journal.

For correspondence please use the following information:

corresponding author: Anna Masek

Institute of Polymer and Dye Technology

Technical University of Lodz

90-924 Lodz, ul Stefanowskiego 12/16, Poland

Tel.: +48 42 631 32 93

Fax: +48 42 636 25 43

Yours sincerely,

Ph. D., D.Sc. Anna Masek

Answer to reviewer #1 comments

Comments and Suggestions for Authors:

  1. From my point of view this review is not thorough enough, and the descriptions of the processes and technologies should be more accurate and detailed. Some statements are not clear and lead to confusion. As an example, authors can find the following very detailed review on very similar topic published in 2019: (Dense CO2 technology: Overview of recent applications for drug processing/ formulation/delivery, Chemical Engineering & Processing: Process Intensification 140 (2019) 64–77, doi.org/10.1016/j.cep.2019.04.009)

Moreover, the following issues should be addressed:

  1. Tittle: When one refers to supercritical conditions, we refer to fluids, not to liquids. (same for Figure 2).
  2. Page 1, line 32: They do not undergo any processes of evaporation, condensation or sublimation. What do you mean? These processes depend on the temperature and pressure conditions.
  3. Page 1, line 35: Is it also possible to find reservoir fluids in supercritical conditions?
  4. Table 1: Fluid, what do you mean by fluid here?
  5. Page 3: Line 85: Check polarization units.
  6. Page 4: Line 123: ‘is the most common’, please elaborate this sentence.
  7. Page 4: Line 125: Can you elaborate more in the fact that the temperature and pressure conditions of the solvent will affect the selectivity of the extraction process? Also some more thorough explanation of the change of the extracted substances from non-polar to polar by increasing pressure. (line 133).
  8. I could not find Table 3 presented in the manuscript.
  9. Page 5, line 162: ‘increases the possibility of a chemical reaction’: Please elaborate more this statement.
  10. SAS: Page 7, line 223: ‘The formation of molecules takes place through crystallization due to the reaction …’. Which reaction are you referring to in the anti-solvent process? Which molecules are formed?
  11. Page 8, line 247: What is the future molecule?
  12. Page 9, line 296: Please define the acronyms used.
  13. Page 9, line 302: What do you mean here by cell nucleation?

Answer: We are grateful for the Reviewer's suggestions and we tried to meet all expectations.

  1. The article written by us was intended to focus mainly on identifying possible changes in the broadly understood polymer industry, these changes would be based on the use of installations using processes in supercritical conditions, and not finding ready-made answers to each change. The process descriptions provided by the process scripts are meant to be more general and less detailed in order to present the essence of the process and method. In our concept, a reader interested in a particular method could go to a specific, more extensive, but also narrower article, in order to learn the details. Hence this form of the article proposed by us. Our approach is also emphasized by the article proposed by the Reviewer "Dense CO2 technology: Overview of recent applications for drug processing / formulation / delivery, Chemical Engineering & Processing: Process Intensification 140 (2019) 64–77, doi.org/10.1016/j.cep. 2019.04.009 "which is more detailed and contains more detailed information about the processes, but only the formation of microcapsules.
  2. Thank you for pointing out the error, it has of course been corrected.
  3. The reason for this opinion was that supercritical fluids do not have phase transitions that are typical and as simple to describe as sublimation, condensation or evaporation. These processes are more complex than for the known liquids, solids or gases.
  4. A very interesting question, I think probably yes, but I have not heard such information.
  5. Thanks for pointing out the error, obviously it was liquids, not liquids at this point in Table 1 - it has been corrected.
  6. In this place should be polarisability (not polarization) and its size is given in the article "Cooper, AI Polymer synthesis and processing using supercritical carbon dioxide. J. Mater. Chem. 2000, 10, 207-234" and is 27.6 × 10- 25 cm3. Thank you for pointing out inaccuracies.
  7. The expression has been changed to be more formal “for the most part”
  8. The article provides additional explanations regarding the influence of sc-CO2 properties and parameters on the solubility of compounds.
  9. Unfortunately, I don't know for what reason, because Table 3 is under line 154.
  10. The expression has been changed to be more formal “increases the selectivity of a chemical reaction”
  11. In the next sentence, after the indicated one, there is a description of the process that explains it "There, the process of diffusion of the anti-solvent to the organic solvent phase takes place, and then the evaporation of the organic solvent into the anti-solvent". This paragraph describes the formation of microcapsules, and these are the particles.
  12. In this case, the terms of the microcapsule. The wording has been changed for clarity.
  13. Acronyms are defined as "Of particular interest are foamed biopolymers such as poly (lactic acid) (PLA) or polycaprolactone (PCL), which can be used to create temporary skeletons in tissue engineering in medicine".
  14. This sentence "After depressurization, CO2 is released, and thermodynamic instability determines the supersaturation of dissolved carbon dioxide in the polymer ma-trix and cell nucleation takes place - as suggested by Karvanja et.al" refers to the nucleation of cells in porous materials such as foams and skeletal structures. It is a term commonly used in the literature of this type of materials, e.g. in the article "Di Maio, E .; Kiran, E. Foaming of polymers with supercritical fluids and perspectives on the current knowledge gaps and challenges. J. Supercrit. Fluids 2018, 134. , 157–166, doi: 10.1016 / j.supflu.2017.11.013. "

The linguistic correction indicated in the review was carried out.

We would like to thank you for any comments that were helpful in improving the text of the article.

Reviewer 3 Report

The review “Application of Liquids in Supercritical Conditions in the Polymer Industry” is an overview on the main supercritical assisted processes for extraction, particles formation and 3-D structure production. This work is particularly useful for the scientific community and also for industry, since supercritical CO2 processes are attracting increasing interest in these last years. For this reason, the publication of this manuscript is recommended; even if, after some revisions.

In particular:

  • 3. Particle formation, micronization and encapsulation.

The state of the art can be enlarged, adding a recent technology assisted by supercritical CO2 for particles/fibers formation: i.e., supercritical assisted electrospray/spinning. See, for instance, the work of Baldino, L., Cardea, S., Reverchon, E. A supercritical CO2 assisted electrohydrodynamic process used to produce microparticles and microfibers of a model polymer, Journal of CO2 Utilization, 2019, 33, pp. 532–540. This new technology overcame the main limitations of the corresponding traditional processes and it is very promising also on industrial scale.

  • 5.1. Foaming.

The review written by Di Maio, E., and Kiran, E. Foaming of polymers with supercritical fluids and perspectives on the current knowledge gaps and challenges. Journal of Supercritical Fluids, 2018, 134, pp. 157–166, can be added and discussed in thi section.

  • In general, some SEM images for each morphology (e.g., micro- and nanoparticles, foams, etc..), can be added, to help the reader in understanding the regular structure of these products.

  • The limits of the corresponding traditional production processes can be highlighted at the beginning of each paragraph, in order to underline the advantages of the supercritical CO2 assisted processes.

  • Some typing errors are present; e.g., CO2.

  • English can be improved.

Author Response

Institute of Polymer and Dye Technology

Technical University of Lodz

90-924 Lodz, ul Stefanowskiego 12/16, Poland

Tel.: +48 42 631 32 23, Fax: +48 42 636 25 43

February 15, 2021

Polymers

Dear Professor,

We are resubmitting our revised paper entitled Application of fluids in Supercritical Conditions in the Polymer Industry by, Karol Tutek, Anna Masek, Anna Kosmalska and Stefan Cichosz with a request to reconsider it for publication in Polymers.

We have carefully considered the Editor and Reviewers' comments. The manuscript was revised exactly according to these comments. The list of responses to the reviewers’ comments and corrections made in the manuscript is attached.

The manuscript has not been previously published, is not currently submitted for review to any other journal, and will not be submitted elsewhere before a decision is made by this journal.

For correspondence please use the following information:

corresponding author: Anna Masek

Institute of Polymer and Dye Technology

Technical University of Lodz

90-924 Lodz, ul Stefanowskiego 12/16, Poland

Tel.: +48 42 631 32 93

Fax: +48 42 636 25 43

Yours sincerely,

Ph. D., D.Sc. Anna Masek

Answer to reviewer #3 comments

Comments and Suggestions for Authors:

The review “Application of Liquids in Supercritical Conditions in the Polymer Industry” is an overview on the main supercritical assisted processes for extraction, particles formation and 3-D structure production. This work is particularly useful for the scientific community and also for industry, since supercritical CO2 processes are attracting increasing interest in these last years. For this reason, the publication of this manuscript is recommended; even if, after some revisions.

In particular:

  • 3. Particle formation, micronization and encapsulation.

The state of the art can be enlarged, adding a recent technology assisted by supercritical CO2 for particles/fibers formation: i.e., supercritical assisted electrospray/spinning. See, for instance, the work of Baldino, L., Cardea, S., Reverchon, E. A supercritical CO2 assisted electrohydrodynamic process used to produce microparticles and microfibers of a model polymer, Journal of CO2 Utilization, 2019, 33, pp. 532–540. This new technology overcame the main limitations of the corresponding traditional processes and it is very promising also on industrial scale.

  • 5.1. Foaming.

The review written by Di Maio, E., and Kiran, E. Foaming of polymers with supercritical fluids and perspectives on the current knowledge gaps and challenges. Journal of Supercritical Fluids, 2018, 134, pp. 157–166, can be added and discussed in thi section.

  • In general, some SEM images for each morphology (e.g., micro- and nanoparticles, foams, etc..), can be added, to help the reader in understanding the regular structure of these products.
  • The limits of the corresponding traditional production processes can be highlighted at the beginning of each paragraph, in order to underline the advantages of the supercritical CO2 assisted processes.
  • Some typing errors are present; e.g., CO2.
  • English can be improved.

Answer: We are grateful for the Reviewer's suggestions and we tried to meet all expectations.

1.Proposed article Baldino, L., Cardea, S., Reverchon, E. A supercritical CO2 assisted electrohydrodynamic process used to produce microparticles and microfibers of a model polymer, Journal of CO2 Utilization, 2019, 33, pp. 532-540 "has been pre-analyzed and quoted in the indicated section" Particle Formation, Micronization and Encapsulation ".

  1. The proposed article "Di Maio, E., and Kiran, E. Foaming of polymers with supercritical fluids and perspectives on the current knowledge gaps and challenges. Journal of Supercritical Fluids, 2018, 134, pp. 157-166" was analyzed and quoted in the indicated "Foaming" section.
  2. Unfortunately, we do not have our own photos, eg SEM, and unfortunately due to formal complications, we do not want to use them from publications of other authors. For this reason, we refrained from adding SEM photos directly in this publication, however, in each section, relevant articles are indicated and cited, which will allow them to be easily found in other, more extensive articles.
  3. The limits of the corresponding traditional production processes were added in some sections where they were absent, in others they were already present, but not necessarily highlighted in each section. Due to the organization of the written text, this has been partially changed where possible without disturbing the continuity of the paragraph structure.
  4. The CO2 errors that were identified and noticed have been corrected.
  5. The linguistic correction indicated in the review was carried out.

We would like to thank you for any comments that have been helpful in improving the text of the article.

Round 2

Reviewer 3 Report

All the modifications proposed by the Reviewer have been performed. The manuscript has been improved.